# MULTICHANNEL GENERATIVE LANGUAGE MODELS

## ABSTRACT

A channel corresponds to a viewpoint or transformation of an underlying meaning. A pair of parallel sentences in English and French express the same underlying meaning but through two separate channels corresponding to their languages. In this work, we present Multichannel Generative Language Models (MGLM), which models the joint distribution over multiple channels, and all its decompositions using a single neural network. MGLM can be trained by feeding it $k$ way parallel-data, bilingual data, or monolingual data across pre-determined channels. MGLM is capable of both conditional generation and unconditional sampling. For conditional generation, the model is given a fully observed channel, and generates the $k - 1$ channels in parallel. In the case of machine translation, this is akin to giving it one source, and the model generates $k - 1$ targets. MGLM can also do partial conditional sampling, where the channels are seeded with prespecified words, and the model is asked to infill the rest. Finally, we can sample from MGLM unconditionally over all $k$ channels. Our experiments on the Multi30K dataset containing English, French, Czech, and German languages suggest that the multitask training with the joint objective leads to improvements in bilingual translations. We provide a quantitative analysis of the quality-diversity trade-offs for different variants of the multichannel model for conditional generation, and a measurement of self-consistency during unconditional generation. We provide qualitative examples for parallel greedy decoding across languages and sampling from the joint distribution of the 4 languages.

## 1 INTRODUCTION

A natural way to consider two parallel sentences in different languages is that each language is expressing the same underlying meaning under a different viewpoint. Each language can be thought of as a transformation that maps an underlying concept into a view that we collectively agree is determined as 'English' or 'French'. Similarly, an image of a cat and the word 'cat' are expressing two views of the same underlying concept. In this case, the image corresponds to a high bandwidth channel and the word 'cat' to a low bandwidth channel. This way of conceptualizing parallel viewpoints naturally leads to the formulation of a fully generative model over each instance, where the transformation corresponds to a particular generation of the underlying view. We define each of these views as a channel. As a concrete example, given a parallel corpus of English and French sentences, English and French become two channels and the corresponding generative model becomes $p(\text{English}, \text{French})$. One key advantage to this formulation is that single model can be trained that can capture the full expressivity of the underlying concept, allowing us to compute conditionals and marginals along with the joint. In the case of parallel sentences, the conditionals correspond to translations from one channel to another while the marginals correspond to standard monolingual language models.

In this work, we present a general framework for modeling the joint distribution $p(\mathbf{x}_1, ..., \mathbf{x}_k)$ over $k$ channels. Our framework marginalizes over all possible factorizations of the joint distribution. Subsequently, this allows our framework to perform, 1) unconditional generation and 2) conditional generation. We harness existing recent work on insertion-based methods that utilize semi-autoregressive models that are permutation-invariant to the joint factorization.

Specifically, we show a proof-of-concept multichannel modeling by extending KERMIT (Chan et al., 2019) to *model the joint distribution over multiple sequence channels*. Specifically, we train KERMIT on the Multi30K (Elliott et al., 2016) machine translation task, consisting of four lan-

guages: English (EN), French (FR), Czech (CS), and German (DE). One advantage of multilingual KERMIT is during inference, we can generate translation for a single target language, or generate translations for $k-1$ languages in parallel in logarithmic time in the token length per language. We illustrate qualitative examples for parallel greedy decoding across languages and sampling from the joint distribution of the 4 languages.

The key contributions in this work are:

1. We present MGLM, a multichannel generative modeling framework. MGLM models the joint distribution $p(\mathbf{x}_1, \ldots, \mathbf{x}_k)$ over $k$ channels.

2. We demonstrate both conditional generation (i.e., machine translation) and unconditional sampling from MGLM.

3. In the case of conditional generation over multiple languages, we show that not only we are competitive in BLEU, but also with significant advantages in inference time and model memory savings.

4. We analyze the Quality-Diversity tradeoff from sampling MGLM and prior work.

We highlight that while we focus on languages as a specific instantiation of a channel, our framework can generalize to any arbitrary specification, such as other types of languages or other modalities.

## 2 BACKGROUND

Traditional autoregressive sequence frameworks (Sutskever et al., 2014; Cho et al., 2014) model the conditional probability $p(\mathbf{y} \mid \mathbf{x})$ of an output sequence $y$ conditioned on the input sequence $x$ with a left-to-right factorization. The model decomposes $p(y \mid x)$ as predicting one output token at time, conditioning on the previously generated output tokens $\mathbf{y}_{<t}$ and the input sequence $\mathbf{x}$:

$$p(\mathbf{y} \mid \mathbf{x}) = \prod_t p(\mathbf{y}_t \mid, \mathbf{y}_{<t}) \tag{1}$$

Recent encoder-decoder models with attention such as Transformer (Vaswani et al., 2017) have been successfully applied to various domains, including machine translation. If we were to apply this left-to-right autoregressive approach towards multichannel modeling, we would require to choose a particular factorization order, such as $p(\mathbf{w}, \mathbf{x}, \mathbf{y}) = p(\mathbf{w})p(\mathbf{x}|\mathbf{w})p(\mathbf{y}|\mathbf{x}, \mathbf{w})$.

Instead of assuming a fixed left-to-right decomposition, recent autoregressive *insertion*-based conditional modeling frameworks (Stern et al., 2019; Welleck et al., 2019; Gu et al., 2019) consider arbitrary factorization of the output sequence by using insertion operation, which predicts both (1) content token $c \in \mathcal{C}$ from the vocabulary, and (2) location $l$ insert, relative to the current partial output $\hat{\mathbf{y}}_t$:

$$p(c, l|\mathbf{x}, \hat{\mathbf{y}}_t) = \text{InsertionTransformer}(\mathbf{x}, \hat{\mathbf{y}}_t) \tag{2}$$

Subsequent work, KERMIT (Chan et al., 2019), simplified the Insertion Transformer model by removing the encoder and only having a decoder, and the trick is to concatenate the original input and output sequence as one single sequence and optimize over all possible factorizations. Consequently, KERMIT is able to model the joint $p(\mathbf{x}, \mathbf{y})$, conditionals $p(\mathbf{x} \mid \mathbf{y})$, $p(\mathbf{y} \mid \mathbf{x})$, as well as the marginals $p(\mathbf{x}), p(\mathbf{y})$.

Unlike with the left-to-right autoregressive approach, the exact computation of the log-likelihood equation 3 is not possible due to the intractable marginalization over the generation order $z$, where $S_n$ denotes the set of all possible permutations on $n$ elements. However, we can lower bound the log-likelihood using Jensen's inequality:

$$\log p(x) = \log \sum_{z \in S_n} p(z)p(\mathbf{x} \mid z) \tag{3}$$

$$\geq \sum_{z \in S_n} p(z) \log p(\mathbf{x} \mid z) \quad =: \mathcal{L}(\mathbf{x}) \tag{4}$$

The loss term can be simplified by changing the summation and careful decomposition of the permutation, leading to:

$$\mathcal{L}(x) = \sum_{z \in S_n} p(z) \log \prod_{i=1}^{n} p((c_i^z, l_i^z) \mid \mathbf{x}_{1:i-1}^{z,i-1})$$

$$= \sum_{i=1}^{n} \sum_{z_{1:i-1}} p(z_{1:i-1}) \sum_{z_i} p(z_i \mid z_{1:i-1}) \log p((c_i^z, l_i^z) \mid \mathbf{x}_{1:i-1}^{z,i-1})$$

Inference can be autoregressive via greedy decoding:

$$(\hat{c}, \hat{l}) = \operatorname*{argmax}_{c,l} p(c, l | \hat{\mathbf{x}}_t), \tag{5}$$

or partially autoregressive via parallel decoding:

$$\hat{c}_l = \operatorname*{argmax}_{c} p(c \mid l, \hat{\mathbf{x}}_t), \tag{6}$$

which is achieved by inserting at all non-finished slots. Stern et al. (2019) has shown that using a binary tree prior for $p(z)$ led to $\approx \log_2 n$ iterations for $n$ token generation.

## 3 MULTICHANNEL GENERATIVE LANGUAGE MODELS

In multichannel generative language modeling, our goal is to learn a generative model given a dataset consisting of a set of sequences $\{\mathbf{x}_1^{(i)}, \dots, \mathbf{x}_k^{(i)}\}_{i=1}^{M}$ from up to $k$ channels, where $\mathbf{x}_k^{(i)} = [x_{j,1}^{(i)}, \dots, x_{j,n}^{(i)}]$ represents a sequence of tokens from the $j$-th channel for the $i$-th example. The resulting MGLM models a joint generative distribution over multiple channels. While there are many possible implementation of Multichannel Generative Language Models, we chose to extended the work of Chan et al. (2019) to investigate applying the KERMIT objective on tasks with more than 2 sequences, in order to learn the joint distribution $p(\mathbf{x}_1, \dots, \mathbf{x}_k)$ over $k$ channel sequences. For example, these channel sequences can denote different languages, such as learning $p(EN, FR, CS, DE)$.

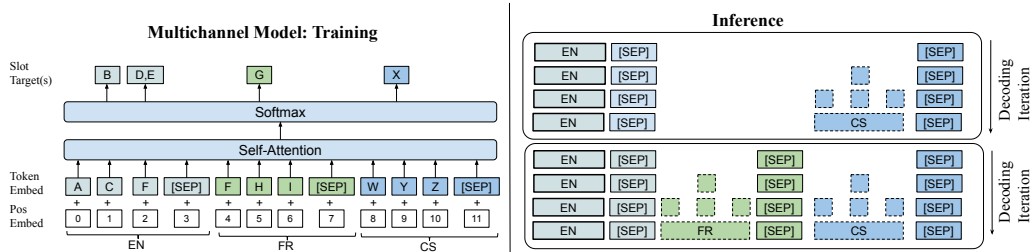

Figure 1: *(Left)* An example multichannel modeling over 3 languages (English, French, Czech), where the model predicts the missing tokens at each location across multiple channels. *(Right)* During inference, MGLM can generate output sequence for a single target language channel (top), or for multiple language channels in parallel (bottom), conditioning on source channel sentence and partial translations of multiple language channels.

We illustrate an example data input consisting of 3 channels in Figure 1 (left). We concatenate the sequences together from all channels for each example, separate by a SEP token. Even with shared vocabulary, each channel results in a different token embedding, via an addition of a channel-specific (learnable) embedding, or simply having a separately learned token embedding per channel. After passing through the dense self-attention layers as in per Transformer architecture, the contextualized representation at each output time step predicts the possible tokens to be inserted to the left of the current input token.

At inference (generation) time, we can generate unconditionally by seeding the canvas with the [SEP] token and predicting the first actual token, or provide as much, or as little, partial/complete sequence in each channel. Figure 1 (right) shows two possible decoding inference modes: a single target language channel (top), or multiple target language channels in parallel (bottom).

## 4 EXPERIMENTS

We experiment on a multilingual dataset to demonstrate that we can learn MGLM. We perform both qualitative and quantitative experiments. We highlight the model's capabilities ranging from conditional generation (i.e., machine translation) to unconditional sampling the joint distribution over multiple languages.

We experiment on the Multi30k (Elliott et al., 2016; 2017; Barrault et al., 2018), a multilingual dataset which consists of 29000 parallel training sentences in English (EN), French (FR), Czech (CS), and German (DE) sentences. We use Multi30k because multiple high quality channels (multilingual translations in this case) is readily available to highlight our framework. We implement MGLM as a base Transformer decoder, without any causal masking, with 6 hidden layers and 1024 dimensional hidden representation. We concatenate all 4 language raw text training examples and use SentencePiece (Kudo & Richardson, 2018) to learn an universal subword unigram (Kudo, 2018) tokenizer with a shared 32K vocabulary size. We follow a similar training set up to BERT (Devlin et al., 2019), using Adam (Kingma & Ba, 2015) optimizer with learning rate of 1e-4, warmup over the first 10% of the total training iterations varying between 10k to 50k iterations. We can train 3 different variants of MGLM by altering the sampling ratio of training data seen by the model:

1. **Bilingual** (e.g., EN → FR). We give the model a fully observed source (e.g., $EN$), and ask the model to infill the target (e.g., $FR$).

2. **Multi-target** (e.g., any 1 → Rest). We give the model a fully observed source (e.g., $EN$), and ask the model to infill the rest of the targets (e.g., $DE, FR, CS$).

3. **Joint**. We ask the model to infill all the targets, consequently we learn a joint distribution over all the languages $p(\text{en}, \text{fr}, \text{de}, \text{cs})$.

### 4.1 TRANSLATION PERFORMANCE

The goal of MGLM is not conditional generation (i.e., machine translation), but nevertheless, we demonstrate its ability to do conditional generation in this section. We report the BLEU scores on the three test sets: `test 2016 Flickr`, `test 2017 Flickr`, `test 2017 MSCOCO`, for different English → {German, French, Czech} translations. We use parallel greedy decoding (Stern et al., 2019; Chan et al., 2019), i.e. inserting to all incomplete slots. Table 1 summarizes the results for English to German and vice versa, respectively. Additional results for English to French, English to Czech, and German to English are shown in Appendix A.2. We observe that the Multi-target models performed similar to slightly better than the bilingual models trained only on a single language pair. This is particularly useful when multiple machine translation targets are desired. We now only need one MGLM model which is competitive to the bidirectional expert models. This implies we only need 1 model for inference over multiple languages, as opposed to $N$ models (i.e., saving substantial memory).

We also observe the full generative joint model has a BLEU gap compared to the bilingual baseline, which is consistent with the findings in Chan et al. (2019). We hypothesize this is due to the joint distribution being a more challenging task. We further hypothesize that in particular, during training the Joint model needs to fantasize additional details when conditioning on partial sequence in each of the channels. This results in fantasizing additional details not present in the original source sentence during translation tasks.

### 4.2 PARALLEL GREEDY DECODING: PARALLEL IN TARGET LANGUAGES

As alluded conceptually in Figure 1 and in the previous section, our KERMIT-based MGLM is also able to perform parallel greedy decoding that is also *parallel in number of target languages*. We illustrate this process in Figure 2. By starting with $K$ initial `[SEP]` tokens for $K$ target output languages, MGLM can decode $K$ target languages that has at most $n$ output tokens per language in $\mathcal{O}(\log n)$, i.e. constant in number of target languages.

We investigate the relative speed up in generating multiple target language outputs in parallel versus generating the targets in series, in terms of wall-clock time and number of decoding iterations. In Figure 3a, we plot the number of decoding iterations taken versus the total output length $N$ for each

| Model | Inference | Test2016 | Test2017 | MSCOCO |
|-------|-----------|----------|----------|--------|
| Bilingual (EN → DE) | EN → DE | 36.14 | 28.32 | 24.15 |
| Bilingual (EN ↔ DE) | EN → DE | **37.08** | 28.69 | 26.11 |
| Multi-target (EN → Rest) | EN → DE | 36.83 | 28.35 | 25.14 |
| | EN → FR,CS,**DE** | 35.41 | **29.69** | 25.64 |
| Multi-target (Any → Rest) | EN → DE | 36.63 | 28.37 | **26.98** |
| | EN → FR,CS,**DE** | 36.51 | 28.53 | 25.84 |
| Joint ($p(EN, FR, CS, DE)$) | EN → DE | 33.06 | 23.42 | 21.39 |
| | EN → FR,CS,**DE** | 32.53 | 23.78 | 20.97 |

Table 1: Multi30k English → German test BLEU.

**Input:** A man sits on a bench holding his dog and looking at the water.
**Parallel Decode:**

**FR:** _Un _homme _est _assis _sur _un _banc , _ten ant _son _chien _et _regardant _l ' eau . [SEP]
**CS:** _Muž _sedí _na _lavičce _a _drží _své ho _psa _a _dívá _se _na _vodu . [SEP]
**DE:** _Ein _Mann _sitzt _auf _einer _Bank _und _hält _seine n _Hund _und _schaut _auf _das _Wasser . [SEP]

**FR:** _Un _homme _est _assis _sur _un _banc , _ten ant _son _chien _et _regardant _l ' eau . [SEP]
**CS:** _Muž _sedí _na _lavičce _a _drží _své ho _psa _a _dívá _se _na _vodu . [SEP]
**DE:** _Ein _Mann _sitzt _auf _einer _Bank _und _hält _seine n _Hund _und _schaut _auf _das _Wasser . [SEP]

**FR:** _Un _homme _est _assis _sur _un _banc , _ten ant _son _chien _et _regardant _l ' eau . [SEP]
**CS:** _Muž _sedí _na _lavičce _a _drží _své ho _psa _a _dívá _se _na _vodu . [SEP]
**DE:** _Ein _Mann _sitzt _auf _einer _Bank _und _hält _seine n _Hund _und _schaut _auf _das _Wasser . [SEP]

**FR:** _Un _homme _est _assis _sur _un _banc , _ten ant _son _chien _et _regardant _l ' eau . [SEP]
**CS:** _Muž _sedí _na _lavičce _a _drží _své ho _psa _a _dívá _se _na _vodu . [SEP]
**DE:** _Ein _Mann _sitzt _auf _einer _Bank _und _hält _seine n _Hund _und _schaut _auf _das _Wasser . [SEP]

**FR:** _Un _homme _est _assis _sur _un _banc , _ten ant _son _chien _et _regardant _l ' eau . [SEP]
**CS:** _Muž _sedí _na _lavičce _a _drží _své ho _psa _a _dívá _se _na _vodu . [SEP]
**DE:** _Ein _Mann _sitzt _auf _einer _Bank _und _hält _seine n _Hund _und _schaut _auf _das _Wasser . [SEP]

Figure 2: Example parallel greedy decode using the Multi-target (Any → Rest) KERMIT model, starting with an English sentence. Blue underlined tokens are the inserted tokens at each iteration, and the gray tokens are the final output that have not been generated yet. Note that the three target languages are generated together in parallel.

sentence in the `test 2016 Flickr` test set, using the Joint KERMIT model when decoding from a single source language to 3 target languages: English → {French, German, Czech}. When performing serial target decoding, we only output the target conditioned on English, i.e. English → French, English → German, English → Czech. We also plot several theoretical bounds: (1) upper bound ($N$) when decoding entirely serially, (2) lower bound $3(\lfloor \log_2(N/3) \rfloor + 2)$ when decoding 3 languages serially but parallel within each language, (3) lower bound $\lfloor \log_2(N/3) \rfloor + 2$, when decoding the 3 target languages in parallel and parallel within each language, and (4) $\lfloor \log_2(N) \rfloor + 2$, if we decode the entire output in parallel as a single sequence. We observe that our model is able to meet the lower bound several times and in many cases decode below the fourth $\lfloor \log_2(N) \rfloor + 2$ bound. Figure 3b compares the wall-clock speed up when decoding targets in parallel vs. in series, with a linear regression line plotted. Our model achieving almost 3 times speed up in wall-clock speed. The parallel targets decoding is bottlenecked by the target language with the longest output sequence. Figure 3c compares the total output length when decoding the targets in series versus in parallel. We observe that there is a linear relationship between the output lengths using the two modes.

## 4.3 CONDITIONAL BILINGUAL GENERATION: QUALITY-DIVERSITY TRADE-OFF

We first evaluated the models on *conditional* generation task by sampling bilingual translations (1 source, 1 target language) for each of the 12 language pair directions. We sample the token and location $(c, l) \sim p(c, l|x, \hat{y})$ from the partial canvas at each iteration, generating 100 hypothesis

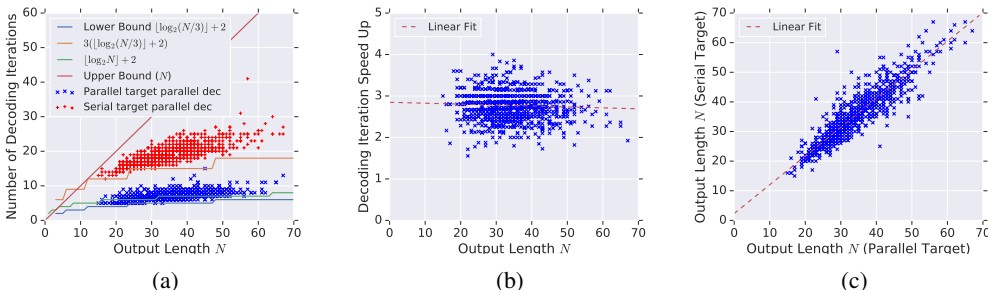

(a)             (b)             (c)

Figure 3: (Left) Number of decoding iterations vs. the output length when decoding each target language serially vs. in parallel, compared to various logarithmic bounds. We have shown that the model is able to achieve close to the theoretical lower bound $\lfloor \log_2(N/k) \rfloor + 2$ where number of target languages $k = 3$. (Middle) Relative wall-clock speed up when using the parallel target languages decoding vs. serial, achieving slightly under 3 times the performance. (Right) Total output length for the 3 target languages when using serial vs parallel target language generation. While not identical, we observe a linear relationship between the output length using the two different modes

translations per source sentence, at softmax temperature $\tau = 0.1, 0.5, 1.0$. At each temperature and model, we computed the *quality* of the generated samples by computing the BLEU Papineni et al. (2002) score between the reference translation and the samples, and the *diversity* by computing the pairwise BLEU between the 100 samples per source, also known as Self-BLEU Zhu et al. (2018). Lower Self-BLEU indicates the higher the diversity as there is less overlap between the samples.

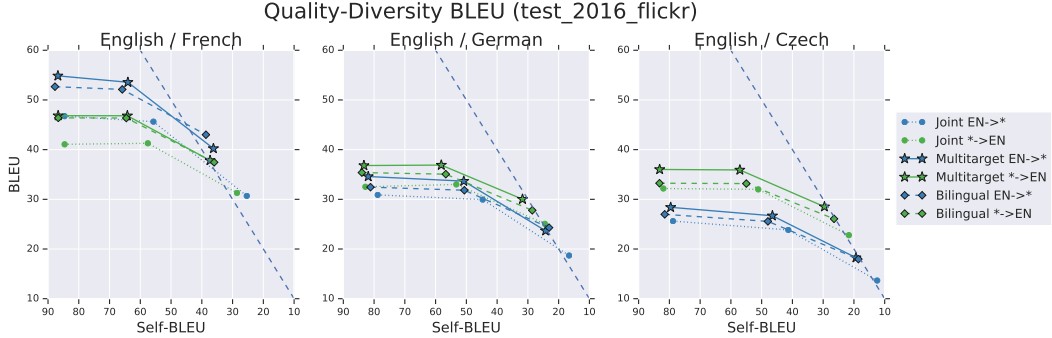

Figure 4: Quality-Diversity BLEU curve for several KERMIT models (bilingual, multitarget, joint) on the Multi30k `text 2016 Flickr` test set. Dotted diagonal line signifies BLEU equals Self-BLEU. Points indicate different temperatures, from 0.1 (low diversity, left in graph) to 1.0 (high diversity, right in graph)

Figure 4 illustrates the Quality-Diversity trade-off for the three models for different translation pairs involving English as one of the language. The top right portion of the graph is the ideal area. We observed that the Multitarget model outperformed the Bilingual model at lower temperature (both higher quality and diversity), and at higher temperature slightly above or below in quality but still higher diversity. Note that only one single Multitarget model was used for all language pair at inference time, while each bilingual model was different for each language pair curve. Therefore, a single Multitarget KERMIT model could outperform specialized bilingual KERMIT models.

## 4.4 PARTIAL CONDITIONING MULTILINGUAL GENERATION

We demonstrate our model's ability to generate infilling for partial conditioning over the multiple channels. To be explicit, we seed each channel with a few (different) words, and sample from the model. We ask the model what text completions would best fit under the model's posterior. Figure 5 highlights several examples for (English, French, German) sentence completion. We took an example from the `test 2016 Flickr` test set and split it into 3 chunks–beginning in English,

middle in French, and ending in German–and sample completion. The model is able to generate a set of diverse, coherent examples.

.......................................................................................................

**English Groundtruth:** A young boy, wearing a chef's hat and apron, is cutting sausages in a kitchen.
**French Groundtruth:** Un jeune garçon, portant une toque et un tablier, coupe des saucisses dans une cuisine.
**German Groundtruth:** Ein kleiner Junge mit Kochmütze und Schürze schneidet in einer Küche Würstchen.

.......................................................................................................

**English Seed:** A young boy,
**French Seed:** portant une toque et un tablier,
**German Seed:** chneidet in einer Küche Würstchen.

.......................................................................................................

**English:** A young boy , wearing a hat , and an apron grilling hotdogs in the kitchen.
**French:** Un jeune garçon portant une toque et un tablier, faisant cuire du citron et des hotdogs dans la cuisine.
**German:** Ein junger Mann trägt eine Mütze und schneidet in einer Küche Würstchen.

**English:** A young boy , wearing a hat and a apron, is in a kitchen , cutting with various foods on it.
**French:** Un jeune garçon, portant une toque et un tablier, est dans une cuisine en projetant des poêles de la nourriture.
**German:** Ein kleiner Junge mit Hut und Schürze schneidet in einer Küche Würstchen.

**English:** A young boy, wearing an orange hat and apron, puts barbecue chicken in a kitchen.
**French:** Un jeune garçon, portant une toque et un tablier, coupant du poulet dans une cuisine.
**German:** Ein kleiner Junge in einer weißen Mütze und mit Schürze schneidet in einer Küche Würstchen glas .

**English:** A young boy, wearing a blue hat and apron, is cooking meat in a kitchen.
**French:** Un petit garçon, portant une toque et un tablier, fait la cuisine dans une cuisine.
**German:** Ein kleiner Junge mit blauer Mütze und schneidet in einer Küche Würstchen.

Figure 5: Partially conditional generation samples drawn from our model. The seed text is shown in gray, with several different in-filling samples from the model in black. The samples show reasonable consistency and diversity across samples.

## 4.5 UNCONDITIONAL MULTILINGUAL GENERATION

We then evaluated the models on *unconditional* multilingual generation task, to generate a sentence each in all 4 languages such that they correspond to each other. For the Joint model, we perform 3 types of sampling: (1) unrestricted, (2) chain, and (3) common cause. For unrestricted, we sampled one (token, location) at each iteration starting from an empty canvas, allowing the model to insert a token in any language, until all slots were marked as completed. In the chain generation, we first restrict to generating English sentence one token at a time, then sampled French, German, and Czech in order, conditioned on the last sentence in the previous language. For common cause, we reuse the same English and French sampled sentences, and generate the German and Czech conditioned on the English sentence (i.e. 3 languages are all conditioned on English).

Given these sets of sentences in 4 languages, for each pair of language direction, we computed a *pseudo* target by using a separately trained (on Multi30k) vanilla Transformer (Vaswani et al., 2017) and performed beam search (size 5) to translate the chosen source language sample. Figure 6 visualizes the pseudo target BLEU score for different source-target language pairs when comparing the Joint model under different types of sampling. The shaded colour represents the difference between the current sampling scheme versus the unrestricted reference. We observe that letting the model sample in unrestricted order was better than either the chain or the common cause sampling.

## 5 RELATED WORK

While we have demonstrated a KERMIT implementation of a MGLM, many other variants of Transformer models contain similar properties. Xia et al. (2019) and He et al. (2018) both consider shared encoder/decoders while KERMIT removes altogether the distinction between the encoder and decoder. XLNet (Yang et al., 2019) also learns over all permutation of the factorization order, in addition to architectural modification for two-stream attention parameterization to resolve ambiguity in the targets. The idea of concatenating *pairs* of source and target sequences from different language channels have been explored by Lample & Conneau (2019). However, unlike the insertion

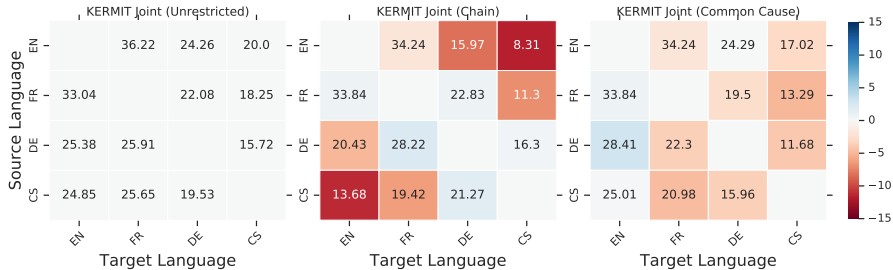

Figure 6: Unconditional multilingual generation Pseudo-Target BLEU for self-consistency when generating sentences in multiple languages. Colour shading indicates the difference compared to the Joint model (unrestricted) generation.

| Model | Language | Generated Sentences |
|---|---|---|
| Joint | English
French
German
Czech | A young man in a blue jacket walking up a mountain.
Un jeune homme en veste bleue descendant une paroi rocheuse en horu.
Ein junger Mann in einer blauen Jacke klettert eine Felswand hoch.
Mladý muž v modré bundě stoupá po horách.
≈"*Young men in blue jackets ascend and climb mountains.*" ✓ |
| Biling. | English
French
German
Czech | Two small white dogs are holding the duck in a fenced yard.
Deux petits chiens blancs tenant un canard dans une cour clôturée.
Zwei kleine weiße Hunde halten eine gelbe Ente in einem eingezäunten Hof.
Dva malí chlapci drží žlutou panou venku u žlutého oploceném nádvoří.
≈"*Two little boys holding a yellow gentleman outside by a yellow fenced courtyard.*" ✗ |

Figure 7: Example unconditional text generation samples from the Joint (top) and chain of Bilingual model (bottom). Note that the Joint model generates one long sequence and we split them into the resulting four sentences in each language here, while Bilingual generate a complete sentence in each language conditioned on previous sentence.

objective, their model is trained through Masked Language Modeling as in BERT (Devlin et al., 2019), and therefore was not readily able to be used for generation.

Evaluation of text generative models remain a challenge (Liu et al., 2016; Novikova et al., 2017). Quality versus diversity plots have been used to compare the trade-off at different output softmax temperatures, as such in Stochastic Beam Search (Kool et al., 2019) which used a simpler $n$-gram diversity instead of Self-BLEU (Zhu et al., 2018). However, we are the first to characterize the Q-D behaviour of insertion based models, versus existing left-to-right language models. Other metrics summarize the quality and diversity trade-off as a single number, such as Frechet BERT Distance (Montahaei et al., 2019) inspired by the FID score (Heusel et al., 2017) used in computer vision, or take into account human evaluation (Hashimoto et al., 2019).

## 6 CONCLUSION

We have demonstrated that a multichannel model implemented with KERMIT can learn a joint distribution over more than two sequences. Furthermore, our multichannel KERMIT model allows for efficient inference of multiple target languages in parallel using a single model. Our work focused on a specific instantiation of channels in the case of languages. However, there are no model limitations that inhibit further generalization to other notion of channels. In future work we aim to consider the addition of multimodal channels, such as images as well as other textual channels, such as paraphrases, premises and hypotheses, as well as questions and answers. Fully generative models still often lag behind purely discriminitive counterparts in terms of performance, but we believe it is crucial to make steps towards other model formulations that have high potential. We also intend to explore the limits on the number of channels that can be considered, such as building generative models over dozens or even hundereds of languages. We hope this initial line of work motivates future research on building generative models of the world.

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

## A  APPENDICES

### A.1  ADDITIONAL QUALITY-DIVERSITY CURVES FOR CONDITIONAL GENERATION

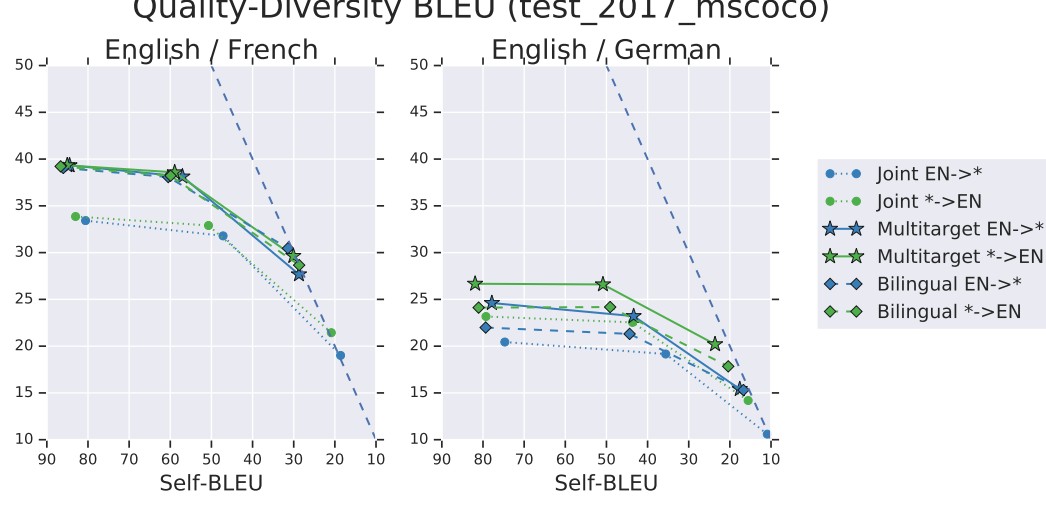

Figure 8: Quality-Diversity BLEU curve for several KERMIT models (bilingual, multitarget, joint) on the Multi30k `text 2017 Flickr` test set. Dotted diagonal line signifies BLEU equals Self-BLEU. Points indicate different temperatures, from 0.1 (low diversity, left in graph) to 1.0 (high diversity, right in graph)

Figure 9: Quality-Diversity BLEU curve for several KERMIT models (bilingual, multitarget, joint) on the Multi30k `text 2017 MSCOCO` test set. Dotted diagonal line signifies BLEU equals Self-BLEU. Points indicate different temperatures, from 0.1 (low diversity, left in graph) to 1.0 (high diversity, right in graph)

## A.2 Additional Multi30K Translation Results

| Model | Inference | Test2016 | Test2017 | MSCOCO |
|---|---|---|---|---|
| Bilingual (EN $\rightarrow$ FR) | EN $\rightarrow$ FR | 58.80 | 50.35 | **42.82** |
| Bilingual (EN $\leftrightarrow$ FR) | EN $\rightarrow$ FR | **59.29** | **52.13** | 42.17 |
| Multi-target (EN $\rightarrow$ Rest) | EN $\rightarrow$ FR | 58.08 | 50.39 | 42.19 |
| | EN $\rightarrow$ **FR**,CS,DE | 58.52 | 50.49 | 41.53 |
| Multi-target (Any $\rightarrow$ Rest) | EN $\rightarrow$ FR | 57.64 | 50.01 | 40.18 |
| | EN $\rightarrow$ **FR**,CS,DE | 57.35 | 48.13 | 39.98 |
| Joint ($p(EN, FR, CS, DE)$) | EN $\rightarrow$ FR | 50.87 | 40.69 | 33.93 |
| | EN $\rightarrow$ **FR**,CS,DE | 48.85 | 39.92 | 33.45 |

Table 2: Multi30k English $\rightarrow$ French test BLEU.

| Model | Inference | Test2016 |
|---|---|---|
| Bilingual (EN $\rightarrow$ CS) | EN $\rightarrow$ CS | 28.58 |
| Bilingual (EN $\leftrightarrow$ CS) | EN $\rightarrow$ CS | 29.03 |
| Multi-target (EN $\rightarrow$ Rest) | EN $\rightarrow$ CS | **30.48** |
| | EN $\rightarrow$ FR,**CS**,DE | 30.15 |
| Multi-target (Any $\rightarrow$ Rest) | EN $\rightarrow$ CS | 30.11 |
| | EN $\rightarrow$ FR,**CS**,DE | 30.11 |
| Joint ($p(EN, FR, CS, DE)$) | EN $\rightarrow$ CS | 26.45 |
| | EN $\rightarrow$ FR,**CS**,DE | 26.35 |

Table 3: Multi30k English $\rightarrow$ Czech test BLEU.

| Model | Inference | Test2016 | Test2017 | MSCOCO |
|---|---|---|---|---|
| Bilingual (DE $\rightarrow$ EN) | DE $\rightarrow$ EN | 39.40 | 34.90 | 27.75 |
| Bilingual (EN $\leftrightarrow$ DE) | DE $\rightarrow$ EN | 40.52 | 35.66 | 28.61 |
| Multi-target (DE $\rightarrow$ Rest) | DE $\rightarrow$ EN | **40.75** | 36.38 | **28.91** |
| | DE $\rightarrow$ **EN**, FR,CS | 39.72 | 35.95 | 28.20 |
| Multi-target (Any $\rightarrow$ Rest) | DE $\rightarrow$ EN | 40.69 | 36.02 | 28.89 |
| | DE $\rightarrow$ **EN**, FR,CS | 39.97 | **37.07** | 28.62 |
| Joint ($p(EN, FR, CS, DE)$) | DE $\rightarrow$ EN | 38.44 | 30.82 | 25.46 |
| | DE $\rightarrow$ **EN**, FR,CS | 36.30 | 29.68 | 24.87 |

Table 4: Multi30k German $\rightarrow$ English test BLEU.

## A.3 Unconditional Sampling Generation

Figure 10 illustrates the serial sampling (one token at a time) from the joint model, every 20 timesteps

| Iterations | Language | Generated Sentence from Joint Model |
|---|---|---|
| 1 | English | |
| | French | |
| | Czech | Mladý |
| | German | |
| 20 | English | |
| | French | descendant |
| | Czech | Mladý muž v modré bundě stoupá po |
| | German | Mann klettert. |
| 40 | English | blue jacket walking up a mountain. |
| | French | veste descendant paroi rocheuse en |
| | Czech | Mladý muž v modré bundě stoupá po horách. |
| | German | Mann klettert. |
| 60 | English | A man blue jacket walking up a mountain. |
| | French | veste bleue descendant une paroi rocheuse en horu. |
| | Czech | Mladý muž v modré bundě stoupá po horách. |
| | German | Mann einer blauen klettert eine hoch. |
| 80 | English | A young man in blue jacket walking up a mountain. |
| | French | veste bleue descendant une paroi rocheuse en horu. |
| | Czech | Mladý muž v modré bundě stoupá po horách. |
| | German | Ein junger Mann in einer blauen Jacke klettert eine Felswand hoch. |
| 96 | English | A young man in a blue jacket walking up a mountain. |
| | French | Un jeune homme en veste bleue descendant une paroi rocheuse en horu. |
| | Czech | Mladý muž v modré bundě stoupá po horách. |
| | German | Ein junger Mann in einer blauen Jacke klettert eine Felswand hoch. |

Figure 10: Example of serial sampling unconditional text generation from the joint $p(EN, FR, CS, DE)$ model, over 96 insertion time steps. Note that the model generates one long sequence and we split them into the resulting four sentences in each language here.

