# OpenReview forum: "Multichannel Generative Language Models"
_ICLR.cc/2020/Conference — Reject_

### Official Review · AnonReviewer2 · 2019-10-22
**Official Blind Review #2**

**Rating:** 3

**Review:**

[Paper summary]
This work is an extension of KERMIT (Chan et al., 2019) to multiple languages and the proposed model is called “multichannel generative language models”. KERMIT is an extension of “Insertion Transformer” (Stern et. al, 2019), a non-autoregressive model that can jointly determine which word and which place the translated words should be inserted. KERMIT shares the encoder and decoder of insertion Transformer, and the source sentence and target sentence are concatenated to train a generative model (also, various loss functions are included). In this work, parallel sentences from more than two languages are concatenated together and fed into KERMIT. Each language is associated with a language embedding. This work demonstrates that a joint distribution p(x1, . . . , xk) over k channels/languages can be properly modeled through a single model. The authors carry out experiments on multi30k dataset.

[Pros] Some discoveries of this work are interesting, including: (1) It is possible to use a single model to translate a sentence into different languages in a non-autoregressive way. (2) The unconditional multilingual generation in Section 4.5 is interesting, especially, the generation order is determined by the model rather than left-to-right.

[Questions]
1.	The authors work on multi30k dataset, which is not a typical dataset for machine translation.
(A)	The dataset and the corresponding information is at https://github.com/multi30k/dataset. The number of words in a sentence is smaller than 15, which is too short for a machine translation. Also, the pattern of sentences is relatively simple.
(B)	For real world application, I am not sure whether it is possible to collect a large amount of k-parallel data where $k>2$. Therefore, the application scenario is limited. What if we have a large amount of bilingual data instead of k-parallel data? How should we leverage the large amount of monolingual data?
2.	For novelty, this is an extension of KERMIT to a multilingual version, which limits the novelty of this wok.
3.	The best results on En->De in Table 1 are inconsistent. On tst16, bilingual en<->de is the best; on tst17, en<->{rest} is the best; on mscoco, any<->rest is the best. In Table 2, seems using bilingual data only is the best choice. This makes me confuse about how to use your proposed method. However,


**Experience Assessment:**

I have published in this field for several years.

**Review Assessment: Checking Correctness Of Derivations And Theory:**

I carefully checked the derivations and theory.

**Review Assessment: Checking Correctness Of Experiments:**

I carefully checked the experiments.

**Review Assessment: Thoroughness In Paper Reading:**

I read the paper thoroughly.

---

> ### Author Response · Authors · 2019-11-15
> **Response to Reviewer #2**
>
> Thank you for taking the time to review our paper. We address your questions below:
>
> 1. Indeed, we agree that Multi30k is not a typical large scale machine translation dataset. However, we chose the Multi30k dataset because it provided us with multiple high quality channels that expresses the same underlying meaning (i.e. the image) under different viewpoint (languages in this case), in order to highlight our approach. While we have not performed experiments with datasets that have partially (m<k) parallel data (such as bilingual pairs or monolingual data as suggested), we believe that our approach should also be able to take advantage of partially parallel data. The partially parallel data should also be weighted by the prior on how often we believe that combination of channels is encountered.
>
> 2. While the Multilingual KERMIT approach can be considered incremental conceptually, we believe that we contributed the novelty in terms of the empirical investigations and characterizing the (unconditional and (partially) conditional) samples from the model. In addition, multilingual KERMIT is only one possible implementation of the proposed MGLM framework, with the hopes that this helps encourage others in the community to pursue this line of generative modeling of multichannel texts.
>
> 3. We will clarify our interpretation of these results in the main paper with more details about the test sets and our hypothesis of the model’s performance. The Flickr test sets are considered “in-domain”, while the MSCOCO are the harder “out-of-domain” where the captions were selected to contain ambiguous verbs, which makes the translation task harder. In those cases, we found that training the model on the more difficult task (i.e. multi-target (any language -> rest) helped with generalization to the MSCOCO test set in the case of English -> German. In the case of English -> French, we hypothesize that the bilingual model performed the best because there is a high mutual information between English and French, such that training on additional languages do not help the model generalize, but rather even distracts the model.

---

### Official Review · AnonReviewer3 · 2019-10-22
**Official Blind Review #3**

**Rating:** 3

**Review:**

This paper proposes a multichannel generative language model (MGLM), which models the joint distribution p(channel_1, ..., channel_k) over k channels. MGLM can be used for both conditional generation (e.g., machine translation) and unconditional sampling. In the experiments, MGLM uses the Multi30k dataset where multiple high quality channels are available, in the form of multilingual translations.

I feel that this paper is not ready for publication at ICLR due to the following major issues:

* Missing important related work: This paper seems unaware of an important related work "Multi-Task Learning for Multiple Language Translation" by Dong et al, ACL 2015. In fact, Dong et al. investigated the problem of learning a machine translation model that can simultaneously translate sentences from one source language to multiple target languages. Although machine translation is just an example of MGLM, Dong et al. is highly relevant to the conditional generation with MGLM, needless to say that they share the same multi-language translation problem domain. Thus, this paper will be much stronger if comparison with important baseline methods is provided.

* Limited novelty: This paper extends Chan et al.'s KERMIT by applying its objective on tasks with more than 2 sequences, in order to learn the joint distribution p(channel_1, ..., channel_k) over k channel sequences. Most of the math in this paper can be found in the original Chan et al.'s paper. The extension to the multichannel case is incremental as it is hard to justify the challenge of such extensions.

Besides, as minor suggestions, it would help readers if more illustrations of Figure 1 (especially the inference part) can be provided.

**Experience Assessment:**

I have published one or two papers in this area.

**Review Assessment: Checking Correctness Of Derivations And Theory:**

I carefully checked the derivations and theory.

**Review Assessment: Checking Correctness Of Experiments:**

I assessed the sensibility of the experiments.

**Review Assessment: Thoroughness In Paper Reading:**

I read the paper at least twice and used my best judgement in assessing the paper.

---

> ### Author Response · Authors · 2019-11-15
> **Response to Reviewer #3**
>
> Thank you for taking the time to review our paper, especially for bringing to our attention an important related work by Dong et al. We will revise the paper to include their work in the related work section and discussion.
>
> One difference with Dong et al. to our approach is that during the multi-target generation, our model’s output at each time step can be conditioned on both the source sentence and the partial translations of all the target languages, while Dong et al.’s model only conditions on the input and the partial translation of the particular target language (in parallel). Our experiment compares the effects of conditioning only on one partial target versus all partial targets inference time.
>
> On novelty: while our specific implementation can be considered incremental from KERMIT, we believe that the task of learning a generative model over several channels is an underexplored direction that is worthwhile to pursue. Despite the vast interest in unconditional generative modeling in images (GANs, VAEs, etc.), we have seen much less interest in the text domain. We also believe that our empirical contributions will help increase interest in this direction by showing what is possible even with using a relatively simple model.

---

### Official Review · AnonReviewer1 · 2019-10-24
**Official Blind Review #1**

**Rating:** 1

**Review:**

This submission belongs to the area of multi-view modelling. In particular, the submission describes construction of multi-view language models that (i) can generate text simultaneously in multiple languages, (ii) can generate text in one or more languages conditioned on text from another language. This submission extends previously proposed KERMIT from two views to more than two views. I believe this paper could be of interest to multi-view modelling/learning community.

Though the original KERMIT approach is very interesting and you application of it to more than two views is also interesting I find the presentation to be poor. In particular I find section 2 to be hard if not impossible to understand without referring to the original paper where the story, equations, nomenclature are much more clearly explained. Even though your extension from two views to multiple is simple I find reliance on a diagram to be a mistake as I find your description not to be very clear. Given that there are no equations to support the reader and that the original equations are not adequate I find it hard to understand Sections 2 and 3. The key experimental result in Table 1 is only briefly commented on despite featuring multiple models with different strength and weaknesses, multiple types of inference. If space is of concern I would suggest removing Figure 2 (or changing input from non-English to English and removing or removing another qualitative table).


**Experience Assessment:**

I have read many papers in this area.

**Review Assessment: Checking Correctness Of Derivations And Theory:**

I carefully checked the derivations and theory.

**Review Assessment: Checking Correctness Of Experiments:**

I carefully checked the experiments.

**Review Assessment: Thoroughness In Paper Reading:**

I read the paper thoroughly.

---

> ### Author Response · Authors · 2019-11-15
> **Response to Reviewer #1**
>
> Thank you for taking the time to review our paper, especially for the constructive feedback on how to improve the clarity of the presentation. We will revise section 2 and 3 to be more clear and self-contained in the future revision, without relying too much on the diagram. We will also expand on the discussion related to Table 1.

---

### Decision · Program_Chairs · 2019-12-19

**Decision:**

Reject

**Comment:**

This paper presents a multi-view generative model which is applied to multilingual text generation. Although all reviewers find the overall approach is important and some results are interesting, the main concern is about the novelty. At the technical level, the proposed method is the extension of the original two-view KERMIT to multiviews, which I have to say incremental. At a higher level, multi-lingual language generation itself is not a very novel idea, and the contribution of the proposed method should be better positioned comparing to related studies. (for example, Dong et al, ACL 2015 as suggested by R#3). Also, some reviewers pointed out the problems in presentation and unconvincing experimental setup. I support the reviewers’ opinions and would like to recommend rejection this time.
I recommend authors to take in the reviewers’ comments and polish the work for the next chance.